# Self-compliant ionic skin by leveraging hierarchical hydrogen bond association

Huating Ye[1], Baohu Wu ®[2], Shengtong Sun ®[1] ✉ & Peiyi Wu ®[1] ✉

Robust interfacial compliance is essential for long-term physiological monitoring via skin-mountable ionic materials. Unfortunately, existing epidermal ionic skins are not compliant and durable enough to accommodate the time-varying deformations of convoluted skin surface, due to an imbalance in viscosity and elasticity. Here we introduce a self-compliant ionic skin that consistently works at the critical gel point state with almost equal viscosity and elasticity over a super-wide frequency range. The material is designed by leveraging hierarchical hydrogen bond association, allowing for the continuous release of polymer strands to create topological entanglements as complementary crosslinks. By embodying properties of rapid stress relaxation, softness, ionic conductivity, self-healability, flaw-insensitivity, self-adhesion, and water-resistance, this ionic skin fosters excellent interfacial compliance with cyclically deforming substrates, and facilitates the acquisition of high-fidelity electrophysiological signals with alleviated motion artifacts. The presented strategy is generalizable and could expand the applicability of epidermal ionic skins to more complex service conditions.

Skin-mounted stretchable epidermal electronics are emerging as an ideal platform for personal health monitoring by collecting timely, high-fidelity electrical signals from human physiological activities, motions, body temperature changes, and so on[1,2]. To achieve this, it is crucial to build a mechanically durable and compliant interface between electronic devices and convoluted skin surface. Traditional rigid electronic components generally rely on ultrathin, low-modulus, and/or adhesive polymer matrices/interlayers to minimize mechanical mismatch[2–9]. In contrast, artificial ionic skins (typically hydrogels, ionogels, eutectogels, and ionic elastomers) by directly mimicking the ion-conducting properties of natural skin have been shown to be an excellent candidate for epidermal electronics[10–13]. Ionic skins have the intrinsic softness and stretchability that are conducive to seamlessly adhering on human skin with spontaneously formed ion-communicating channels[14–16]. In addition, the customizable network designs of ionic skins, which are based on various dynamic interactions, have spawned a wide range of skin-like mechanical properties such as strain-stiffening, damping, crack-tolerant, and anti-fatigue behavior[17–20].

Although promising strides have been made in the development of ionic skins, previous studies mainly focused on the compatibility of static mechanical properties (e.g. softness, stretchability, adhesion) with skin. Nevertheless, this is insufficient to meet the critical requirement of epidermal electronics for long-term use under dynamic conditions. Human skin is constantly deforming in a variety of ways (bending, bulging, jiggling, and stretching) and at time-varying strains (0-50%) and frequencies (0-50 Hz). Without dynamic compliance, the stress concentration of ionic skins will inevitably accumulate during the repetitive deformation of human skin, which would incur eventual interfacial failure and signal artifacts. Unfortunately, current ionic skins are typically dominated by either viscosity or elasticity (Fig. 1a). Liquid-like viscous ionic skins allow for easy spreading and compliance on the deformed human skin, yet their poor shape retention makes them unsuitable for durable signal recording[13,21–23]. Elastic ionic skins durably recover from deformation, but they poorly adapt to stretched skin microstructures, especially in prolonged deformations[15,17,18]. To date, it remains a major limitation for epidermal

[1]State Key Laboratory for Modification of Chemical Fibers and Polymer Materials, College of Chemistry and Chemical Engineering & Center for Advanced Low-dimension Materials, Donghua University, Shanghai 201620, China. [2]Jülich Centre for Neutron Science (JCNS) at Heinz Maier-Leibnitz Zentrum (MLZ) Forschungszentrum Jülich, Lichtenbergstr. 1, 85748 Garching, Germany. ✉e-mail: shengtongsun@dhu.edu.cn; wupeiyi@dhu.edu.cn

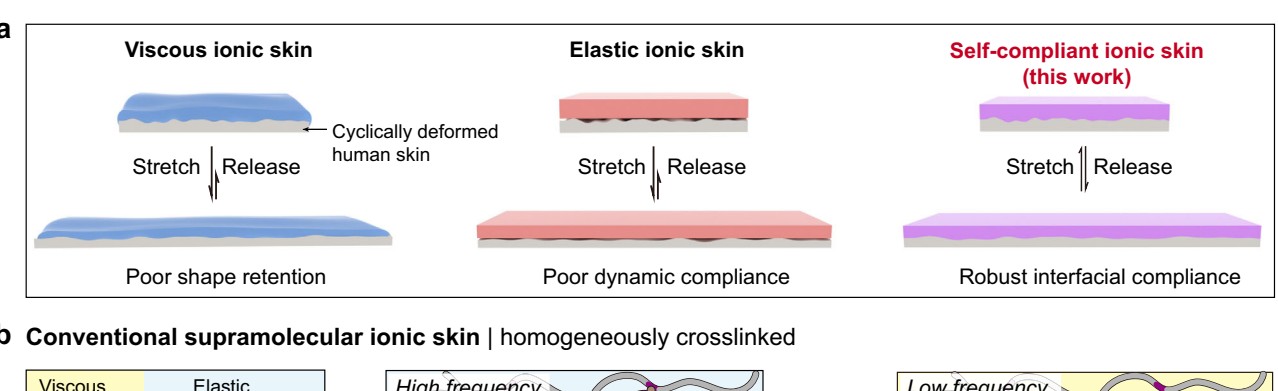

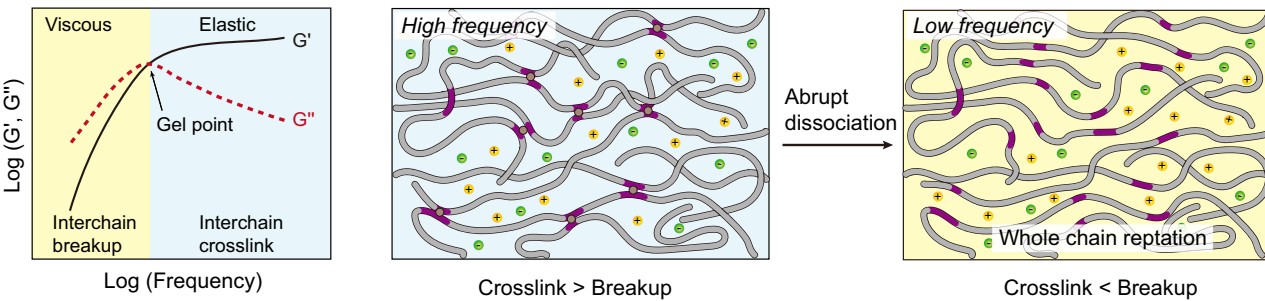

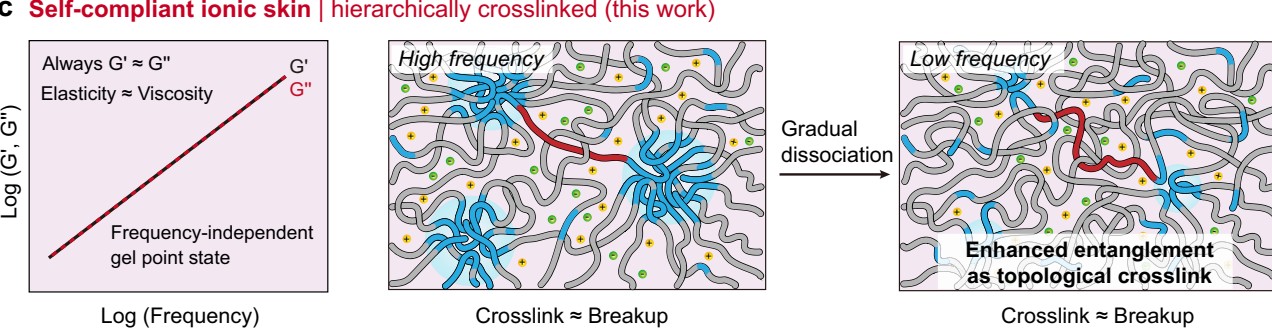

**Fig. 1 | Working mechanism of self-compliant ionic skin. a** Schematic dynamic compliance of viscous, elastic, and self-compliant ionic skins. **b** Typical rheological behavior and working mechanism of conventional supramolecular ionic skin with a homogeneously crosslinked structure. With decreasing frequency, the abrupt dissociation of physical interactions occurs leading to the sharp transition between viscous and elastic states. **c** Frequency-independent gel point state and working mechanism of self-compliant ionic skin with a hierarchically crosslinked structure. With decreasing frequency, the gradual dissociation of physical interactions takes place, resulting in simultaneous interchain breakup and enhanced entanglement as topological crosslink.

ionic skins to achieve the balanced viscosity and elasticity for dynamic compliance and durable shape recovery, respectively.

It is noted that natural skins and tissues are all viscoelastic and exhibit proper stress relaxation for self-regulated movement and growing. For example, the subcutaneous adipose tissue interlayer has high stress relaxation to enable the adaptive conformality between moving muscles and skins[19,24]. The growing liver and brain also exhibit remarkable stress relaxation that helps to match the surrounding tissues[25]. Synthetic supramolecular polymer networks can also exhibit viscoelastic properties, and the maximum stress relaxation generally occurs at the so-called gel point state, where the storage modulus ($G'$, which denotes elasticity) and loss modulus ($G''$, which denotes viscosity) are equal. This means that a material close to the gel point (with the loss factor $\tan \delta = G''/G' \approx 1$) can not only dissipate strain energy for dynamic compliance but also durably recover to avoid free flow[26,27].

Based on these findings, in this paper, we develop a self-compliant ionic skin that works consistently at the critical gel point state with almost equal viscosity and elasticity across a super-wide frequency range ($10^{-11} – 500$ Hz). This is in stark contrast to conventional supramolecular ionic skins, which have only one gel point at a fixed frequency with either viscosity or elasticity being dominant. Such a frequency-independent gel point state is particularly achieved by

leveraging phase-separated hierarchical hydrogen bond (H-bond) association. Mechanistically, with decreasing frequency, the released polymer strands from gradual H-bond dissociation simultaneously enhance chain entanglements, which can act as complementary topological crosslinks to maintain the viscosity-elasticity equilibrium. We use a model ionogel made from an associative polymer (poly(butyl acrylate-*co*-methacrylic acid), abbreviated as P(BA-*co*-MAA)) to prove this concept, but it can also be easily extended to other similar systems. The optimal ionic skin is highly transparent (-92%), ultrasoft (modulus -14 kPa), super-stretchable (2900%), self-healable (-97% efficiency), flaw-insensitive (fractocohesive length -1.2 cm), self-adhesive (-370 J m$^{-2}$ on porcine skin), and water-resistant. All these intriguing properties contribute to its durable self-compliance with the repetitively deformed skin-mimetic substrates, which cannot be readily realized by existing elastic adhesive materials. Finally, we demonstrate the use of this self-compliant ionic skin for epidermal electromyography (EMG) and electrocardiography (ECG) monitoring, which both show alleviated vibration and motion artifacts.

## Results

### Working mechanism of self-compliant ionic skin

Conventional supramolecular ionic skins are generally homogeneously crosslinked with reversible physical interactions (Fig. 1b).

The rheological state of these materials depends on the competition between the thermal energy ($kT = 2.5$ kJ mol$^{-1}$ at ambient temperature and pressure) which breaks interchain crosslinks, and the association energy which favors the formation of interchain crosslinks[28]. At lower frequencies (equivalent to higher temperatures according to the principle of time-temperature equivalence), interchain breakup dominates. This allows for whole chain reptation, and as a result, the material appears as a viscous sol ($G' < G''$). On the contrary, at higher frequencies, interchain crosslink via physical interactions prevails and the material appears as an elastic gel ($G' > G''$). At the gel point ($G' \approx G''$), the roles of interchain crosslink and breakup reach an equilibrium. This corresponds to an average of one association per chain under the assumption of random placement of associative groups[28]. Since the binding strength of physical interactions in the homogeneously crosslinked network is usually narrowly distributed, abrupt chain dissociation would occur across a fixed frequency, resulting in a sharp transition between the viscous and elastic states and only a single gel point.

We believe that if the equilibrium between interchain crosslink and breakup can be maintained over a wide frequency range, a self-compliant ionic skin working consistently at the gel point state can be thus obtained. We here propose a solution by means of hierarchical crosslinking design with phase-separated nanostructure (Fig. 1c). Such a hierarchical structure can greatly broaden the distribution of the binding strength of physical interactions. In general, the larger the formed aggregates, the stronger the binding strength due to cooperative motions, and the longer the lifetime of physical crosslinks[29,30]. With decreasing frequency, gradual chain dissociation would take place, since the larger aggregates are more resistant to dissociation and remain intact. As a consequence, despite the reduction of the number density of associative crosslinks, longer network strands will be released to enhance chain entanglements, which are known to act as topological crosslinks[31–33]. Therefore, by delicately adjusting the number density of associative groups as well as the phase-separated structure of supramolecular network, the complementary effect between associative crosslinks and entanglement crosslinks shall maintain the material consistently at the gel point state. Note that, the entanglement crosslinks are slippery and not as strong as the associative crosslinks, and thus the moduli of the material would still continuously decrease with decreasing frequency.

## Fabrication and optimization of self-compliant ionic skin

To prove the above hypothesis, we chose a typical ionogel as the proof-of-concept system for self-compliant ionic skin. The ionogel was synthesized via a one-step photo-induced copolymerization process, which consists of an associative copolymer (P(BA-co-MAA)) as the supramolecular network and an ionic liquid as the ion-conducting agent. The ionic liquid, butyltrimethylammonium bis(trifluoromethylsulfonyl)imide (abbreviated as [N$_{4111}$][TFSI])) was particularly used owing to its high hydrophobicity, low toxicity, as well as its structural similarity (cation) to PBA[16]. The random copolymerization of BA and MAA was expected in consideration of their reactivity ratios ($r_{BA} = 0.35$, $r_{MAA} = 1.31$)[34]. In the designed ionogel, PBA chains are compatible with [N$_{4111}$][TFSI] (i.e., ionophilic) while PMAA chains are incompatible with both PBA and [N$_{4111}$][TFSI] (i.e., ionophobic) (Fig. 2a). Hierarchical H-bond association can thus be expected, as the COOH groups of PMAA are known to form H-bond assemblies from cyclic dimers to linear oligomers with a broad binding strength distribution[35,36]. It is also noted that, we chose PMAA rather than polyacrylic acid (PAA) as the COOH-bearing polymer because the hydrophobic α-methyl groups contribute to much stronger H-bond associations and thus higher stability over frequency changes[37].

We carefully adjusted the molar content of MAA with respect to BA from 0 to 0.3 to regulate the hierarchical nanostructure and balance the viscoelasticity of the resulting ionogels. At the MAA contents less than 0.05, the number density of H-bond association was not high enough to shape the ionogel, which appeared as a viscous liquid like neat PBA ionogel (Fig. 2b). With further increasing MAA contents, the ionogels shaped well at ambient conditions, and their corresponding appearance transformed gradually from transparent to opaque. Dramatic transmittance reduction took place at the MAA contents of 0.1 and 0.3, suggesting the formation of larger PMAA-rich H-bond aggregates that strongly scattered incident visible light (Fig. 2b and Supplementary Fig. 1). The growth of H-bond aggregates with increasing MAA contents can be clearly observed by scanning electron microscopy (SEM) (Fig. 2b). P(BA-co-MAA) ionogels with the MAA contents lower than 0.05 showed a uniform and smooth cross-sectional morphology indicating no apparent phase separation. Hierarchical H-bond aggregation emerged at the MAA content of 0.05 and gradually grew with increasing sizes by further increasing MAA contents. Apparent bicontinuous structure was observed at the MAA content of 0.3, suggesting the coalescence of H-bond aggregates at this composition[38].

Small-angle X-ray scattering (SAXS) and low-field $^1$H NMR were employed to evaluate the structural changes of ionogels at the nanoscale. The SAXS results revealed a multi-level structural evolution across the samples (Fig. 2c). The slight inflection of the scattering profiles at $q < 0.2$ nm$^{-1}$ appeared for the ionogels with MAA contents from 0.05 to 0.3, suggesting the emergence of structural inhomogeneities induced by phase separation. This trend became prominent with increasing MAA contents and culminated at the MAA content of 0.3 in the formation of a bicontinuous phase structure. Correspondingly, the calculated average radii of gyration of nanophases increased from ~56 nm at the MAA content of 0.05 to ~85 nm at the MAA content of 0.1. Low-field $^1$H NMR is powerful to monitor the activity changes of hydrogen atoms via spin-spin relaxation time ($T_2$)[39]. As shown in Fig. 2d, the $T_2$ peak assigned to PBA chains first shifted to lower relaxation times with increasing MAA contents from 0 to 0.05, and then remained at the same position till the formation of a bicontinuous structure at the MAA content of 0.3. Obviously, the MAA content of 0.05 is supposed to be the critical point for hierarchical H-bond association, which started to lock the free flow of PBA chains.

The hierarchical H-bond association-induced nanostructure evolution also led to the remarkable changes of ionogels in electrical and mechanical properties. Although the glass transition temperatures of the ionogels were not significantly influenced by MAA contents (about -50 °C, Supplementary Fig. 2), the calculated ionic conductivities decreased from $4.4 \times 10^{-4}$ S m$^{-1}$ for neat PBA ionogel to $4.1 \times 10^{-5}$ S m$^{-1}$ for P(BA-co-MAA$_{0.3}$) ionogel (Fig. 2e). This trend corresponds to the increased crosslinking density with increasing MAA contents which limited the ion migration across polymer network. MAA contents also play an important role in the mechanical properties of the resulting materials (Fig. 2f; see calculated Young's moduli in Fig. 2g). The ionogels with MAA molar contents less than 0.05 displayed strain-softening behavior due to their viscous nature (Supplementary Fig. 3). Good elasticity with superhigh stretchability (2200%-2900%) was observed for the ionogels with MAA contents ranging from 0.05 to 0.1. Further increasing MAA content to 0.3 or even larger made the ionogel stiffer yet less ductile due to the formation of bicontinuous structure (Supplementary Fig. 4).

The self-compliance of ionic skin strongly relies on the balanced viscosity and elasticity. We measured the frequency-sweep rheological behavior of P(BA-co-MAA) ionogels with different MAA contents at room temperature (see all the data in Supplementary Fig. 5). The tan $\delta$ value at 0.1 Hz was chosen as the evaluating factor to optimize the recipe. As presented in Fig. 2g, the transition from viscosity (tan $\delta > 1$; for neat PBA ionogel, tan $\delta = 11.5$) to elasticity (tan $\delta < 1$) occurred at the MAA content of 0.05, in accord with previous results. Note that, the tan $\delta$ value of P(BA-co-MAA$_{0.05}$) ionogel (tan $\delta = 0.93$) was delicately

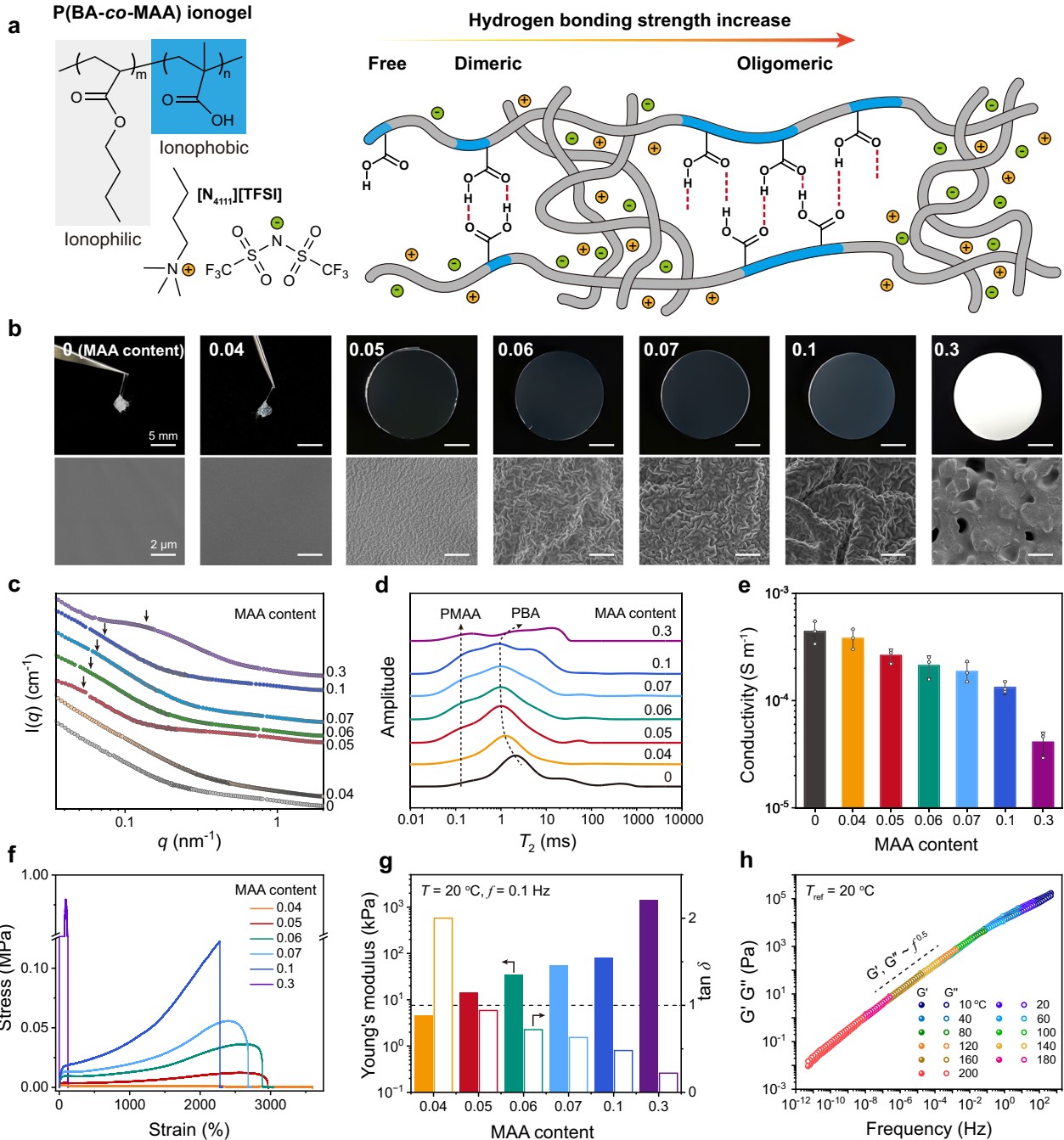

**Fig. 2 | Fabrication and optimization of self-compliant ionic skin. a** Schematic illustration of the chemical structure of P(BA-*co*-MAA) ionogel and the distribution of hydrogen bonding strength. **b** Photographs (upper) and SEM images (bottom) of the ionogels with increasing MAA molar contents. **c** Corresponding SAXS profiles (arrows indicate the presence of phase separation). **d** Low-field ¹H NMR curves. **e** Ionic conductivities. **f** Tensile stress-strain curves (strain rate: 0.05 s⁻¹).

**g** Calculated Young's moduli and tan $\delta$ values measured at 0.1 Hz and 20 °C (the dashed line denotes tan $\delta$ = 1). **h** Time-temperature superposition rheological master curves of P(BA-*co*-MAA$_{0.05}$) ionogel at the reference temperature of 20 °C. Data are presented as the mean values ± SD, $n$ = 3 independent samples. Source data are provided as a Source Data file.

adjusted to be slightly lower than 1, which guarantees good elastic recovery for shape retention (Supplementary Fig. 6).

We further plotted the rheological master curves of P(BA-*co*-MAA$_{0.05}$) ionogel following the time-temperature superposition principle at the reference temperature of 20 °C (Fig. 2h, see original data in Supplementary Fig. 7). In contrast to typical viscoelastic polymers which have distinct viscous and elastic (or rubbery) regions separated by a single gel point (Fig. 1b), P(BA-*co*-MAA$_{0.05}$) ionogel exhibited a constant gel point state ($G' \approx G''$) over a super-wide frequency range

(10⁻¹¹ – 500 Hz, spanning 13 orders of magnitude). We highlight that, although a few previous studies have reported hydrogels and elastomers with close $G'$ and $G''$ at certain frequencies[26,27,39–41], none of these materials exhibited such a wide-range gel point state which is almost independent of frequency (see comparison in Supplementary Table 1). The rather low frequency limit of the gel point state of the ionogel not only avoids free flow for long-term uses but also enables potential uses at high temperatures till 200 °C for soft robotics and artificial prosthetics. The temperature-sweep rheological curves of the ionogel

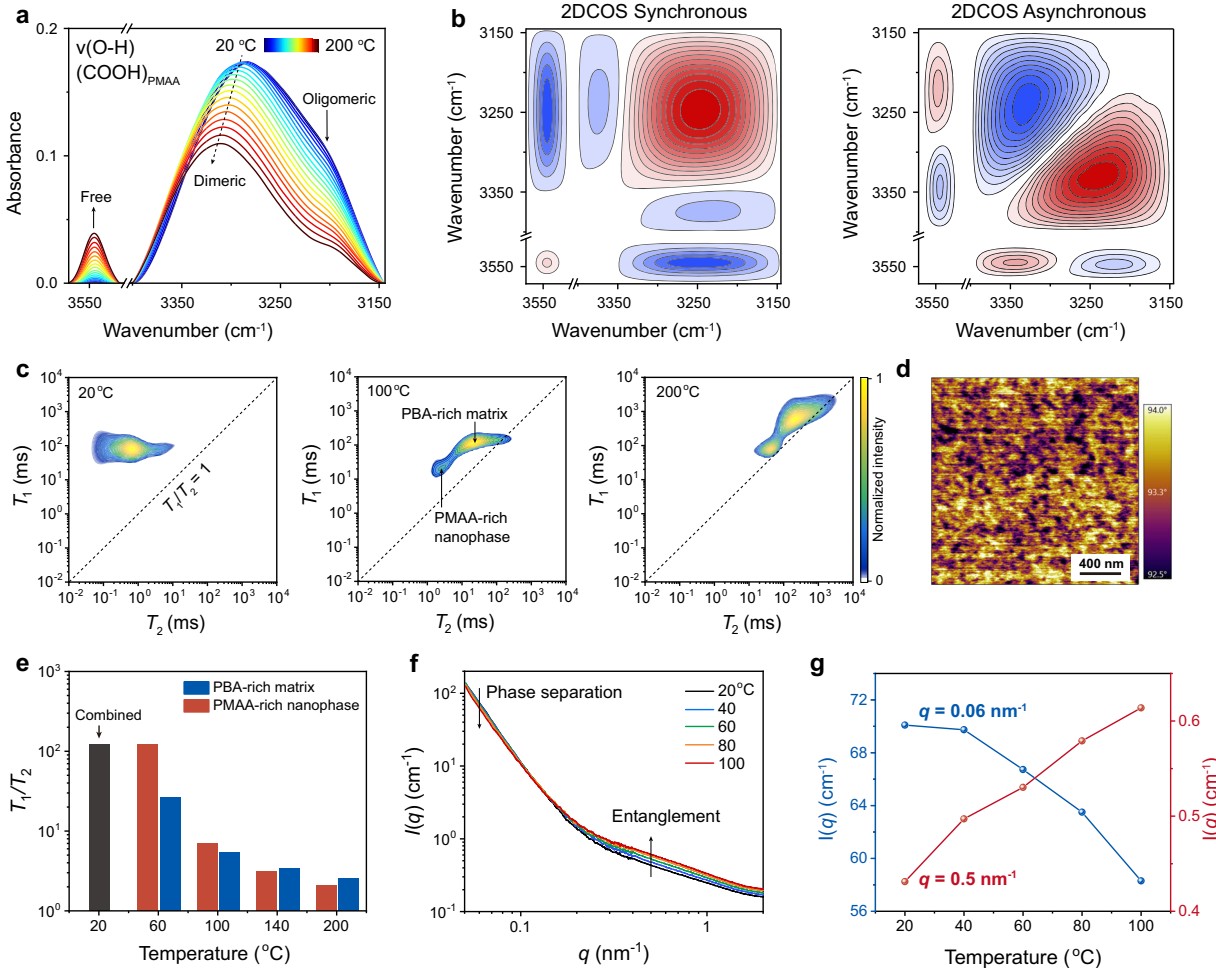

**Fig. 3 | Mechanism analysis for the gel point state of P(BA-*co*-MAA) ionogel.**
**a** Temperature-variable IR spectra of P(BA-*co*-MAA$_{0.05}$) ionogel upon heating from 20 to 200 °C (interval: 10 °C). **b** 2DCOS synchronous and asynchronous spectra generated from (a). In 2DCOS spectra, red colors represent positive intensities, while blue colors represent negative intensities. **c** 2D low-field $^1$H NMR spectra at 20, 100, and 200 °C, respectively. **d** AFM phase image of the ionogel at 20 °C. **e** Temperature-dependent $T_1/T_2$ ratios of PBA-rich matrix and PMAA-rich nano-phase, respectively. **f** Temperature-variable SAXS profiles. **g** Temperature-dependent SAXS intensity changes at $q = 0.06$ and $0.5$ nm$^{-1}$, respectively.

further consolidated this conclusion (Supplementary Fig. 8). It is also noted that, the emergence of a very small entanglement plateau at higher frequencies suggests that the viscoelastic region of P(BA-*co*-MAA$_{0.05}$) ionogel with $G' \approx G''$ is indeed the gel point state, rather than the glass transition state (which appears at even higher frequencies than the entanglement plateau). The whole rheological master curves followed a scaling of $G'$, $G'' \sim f^{0.5}$, which can be attributed to the associative Rouse mode from reversible crosslinks[42,43]. The calculated apparent association energy of 124.7 kJ mol$^{-1}$ (Supplementary Fig. 9) is much higher than the dimerization energy of COOH groups (-25-40 kJ mol$^{-1}$)[44], further consolidating the presence of hierarchical H-bond association in the ionogel.

All the above characterizations pointed to an optimal recipe of P(BA-*co*-MAA$_{0.05}$) ionogel for the desired self-compliance, which showed a very low Young's modulus (14 kPa), superb stretchability (2900%), high transparency (-92%), adequate ionic conductivity (2.6 × 10$^{-4}$ S m$^{-1}$), and more intriguingly, the gel point state across a super-wide frequency range. Thereafter, without otherwise stated, all the ionogels in the following studies are referred to the sample of P(BA-*co*-MAA$_{0.05}$) ionogel. We further emphasize that our design is also generalizable by replacing BA with other monomers like methyl acrylate (MA), ethyl acrylate (EA), isobutyl acrylate (iBA), and 2-metheoxyethyl acrylate (MEA), which all showed the gel point state

over a super-wide frequency range (Supplementary Fig. 10). If ionic liquid was removed from the ionogel, the resulting P(BA-*co*-MAA) elastomer exhibited still gel point state in a super-wide frequency range (Supplementary Fig. 11). However, replacing MAA by AA or sodium methacrylate (MAANa) was not able to reproduce the wide-frequency-range gel point state, probably due to the weakened binding strength that could not maintain stable association at lower frequencies (Supplementary Fig. 12). These control experiments suggest that, designing gel-point-state materials should mainly focus on the proper selection of associating components, which are the key to governing whole chain dynamics.

**Mechanism discussion for gel point state**

To support the proposed mechanistic relationship between hierarchical H-bond association and the gel point state in the P(BA-*co*-MAA) ionogel (Fig. 1c), we performed a few temperature-dependent characterizations, since temperature changes are equivalent to frequency variations in viscoelastic polymer systems. We first collected temperature-variable transmission IR spectra from 20 to 200 °C to evaluate the heat-induced H-bond evolution of P(BA-*co*-MAA) ionogel. Here we focused on the O-H stretching ($\nu$(O-H)) region of COOH groups in PMAA moieties. As shown in Fig. 3a, a diversity of interaction forms can be observed mainly involving free COOH, dimeric H-bond,

and oligomeric H-bond with descending wavenumbers. Noteworthily, the very wide spectral range for H-bond forms from 3400 cm$^{-1}$ to 3145 cm$^{-1}$ is a clear indication of the broadness of interactions with varying strengths[35]. With increasing temperatures, the spectral intensities of dimeric and oligomeric H-bonds were gradually reduced while the spectral intensity of free COOH increased (the peak for dimeric H-bond also shifted to higher wavenumbers), suggesting the heat-induced H-bond disassociation among PMAA moieties. The transformation of COOH groups from H-bonded to free forms was further validated by the deconvoluted analysis at different temperatures (Supplementary Fig. 13).

Two-dimensional correlation spectroscopy (2DCOS) was further employed to extract more subtle information about interaction changes in the ionogel (Fig. 3b). Two sets of spectra, synchronous and asynchronous, were generated to reflect the synchronized and unsynchronized changes of spectral intensities at two given wavenumbers, respectively[17,18,37,45]. Notably, two wavenumbers ascribed to the respective strong and weak forms of H-bonds have been clearly discerned by the asynchronous spectrum for both oligomeric H-bond (3220 cm$^{-1}$/3240 cm$^{-1}$) and dimeric H-bond (3330 cm$^{-1}$/3342 cm$^{-1}$). In light of Noda's judging rule, the responsive order of the different forms of COOH groups to temperature increase is 3220 cm$^{-1}$ → 3240 cm$^{-1}$ → 3545 cm$^{-1}$ → 3330 cm$^{-1}$ → 3342 cm$^{-1}$ (→ means prior to or earlier than; see determination details in Supplementary Table 2), i.e., $v$(O-H) (oligomeric H-bond) (strong) → $v$(O-H) (oligomeric H-bond) (weak) → $v$(O-H) (free COOH) → $v$(O-H) (dimeric H-bond) (strong) → $v$(O-H) (dimeric H-bond) (weak). The earliest response of oligomeric H-bonds revealed that the heat-induced viscoelastic changes of the ionogel are indeed driven by hierarchical H-bond dissociation.

Temperature-variable 2D low-field $^{1}$H NMR was employed to investigate the mobility changes of different species in the ionogel (Fig. 3c and Supplementary Fig. 14). In the 2D $T_1$-$T_2$ map ($T_1$: spin-lattice relaxation time; $T_2$: spin-spin relaxation time), only one cross-peak was observed at room temperature, demonstrating the cooperative motions of all the nanostructures. Increasing temperature clearly differentiated the ionogel into two condensed nanostructures, which can be ascribed to H-bonded PMAA-rich nanophase and PBA-rich matrix, locating at lower and higher $T_2$, respectively. The presence of H-bonded PMAA-rich nanophases has been clearly observed by atomic force microscope (AFM) with sizes between 50 and 100 nm (Fig. 3d). Moreover, the $T_1/T_2$ ratios of the two cross-peaks in 2D low-field $^{1}$H NMR were additionally plotted as a function of temperature (Fig. 3e), which are a measure of molecular mobilities[46]. Generally, a lower $T_1/T_2$ ratio means a higher mobility (the diagonal line with $T_1/T_2 = 1$ corresponds to the completely mobile liquid state). Obviously, with heat-induced H-bond dissociation, both the PMAA-rich nanophase and PBA-rich matrix became more mobile. At temperatures higher than 100 °C, the mobilities of these two components were almost the same, corresponding to the largely weakened structural inhomogeneity due to the reduction of the number density of H-bond associations.

To further support the enhanced chain entanglement by H-bond dissociation (Fig. 1c), we measured the number-average molecular weight of as-prepared P(BA-co-MAA) by gel permeation chromatography (GPC) to be about $1.1 \times 10^5$ g mol$^{-1}$ (Supplementary Fig. 15). Such a molecular weight is much higher than the entanglement molecular mass of linear PBA ($\approx 2 \times 10^4$ g mol$^{-1}$)[33], suggesting the occurrence of sufficient chain entanglement in the ionogel. Temperature-variable SAXS analysis was then performed to evaluate the heat-induced nanostructure changes of the ionogel. Interestingly, SAXS curves exhibited a binary intensity change at the small and high $q$ regimes (Fig. 3f and g). At the high $q$ regime >0.11 nm$^{-1}$, the scattering intensity raised as the temperature increased, indicating the enhancement of polymer chain entanglement. Conversely, the low $q$ scattering intensity diminished with rising temperature, which implies again the

weakening of structural inhomogeneity associated with phase separation.

Altogether, the above temperature-variable 1D/2D IR, low-field $^{1}$H NMR, and SAXS findings clearly revealed the heat-induced hierarchical H-bond dissociation along with the increase of chain mobility and entanglement in the P(BA-co-MAA) ionogel. This proof-of-concept study consolidates our hypothesis of hierarchical crosslinking strategy in designing gel-point-state materials, and importantly, highlights the critical role of gradual H-bond dissociation in enhancing chain entanglements as complementary topological crosslinks.

## Flaw-insensitive, self-healing, self-adhesive, and self-compliant properties

We further demonstrate that the hierarchical structure and the gel-point-state rheological behavior of P(BA-co-MAA) ionogel impart it with a few fascinating properties including flaw-insensitivity, self-healing, self-adhesion, and durable self-compliance. The presence of high-modulus H-bond nanoaggregates helps with crack blunting by significantly increasing the energy to fracture the material[37]. According to the pure shear test, a notch (~1/5 of sample width) was created by a sharp blade, and tensile test was then performed to examine the flaw-tolerant performance. The critical strain of the notched ionogel was 2850%, approximate to the maximum elongation of the unnotched one, suggesting that the ionogel is almost insensitive to flaws (Fig. 4a and Supplementary Movie 1). The fracture energy of the ionogel was calculated to be 2.1 kJ m$^{-2}$, and the fractocohesive length was as high as 1.2 cm, below which the ultimate material properties are independent of crack length. It is highlighted that the fractocohesive length of the ionogel is comparable to many soft tissues such as bovine pericardium and rhinoceros dermal (~1 cm) and larger than most synthetic hydrogels and elastomers[47].

Moreover, akin to natural skins, P(BA-co-MAA) ionogel is also self-healable at room temperature without the assistance of heat or solvent owing to its supramolecular nature and highly dynamic chain association. As depicted in Fig. 4b, the scars on the ionogel film almost disappeared in 24 h at room temperature. We quantitatively evaluated the self-healing efficiency of the ionogel by tensile stress–strain curves. When healed at room temperature for 24 h, the film could recover the fracture strain of ~2800%, corresponding to a superhigh healing efficiency of 97% (Fig. 4c). The highly dynamic nature of the ionogel even allowed solution reprocessibility, and the recast ionogel from its solution in ethyl acetate reproduced both the mechanical and gel-point-state behavior (Supplementary Fig. 16).

Robust interfacial adhesion is important for epidermal electronics to intimately contact human skin. To evaluate the adhesion strength between the ionogel and various substrates, 90° peeling tests were performed. In the air, the measured peeling adhesion strengths or interfacial toughnesses on different substrates (copper, thermoplastic elastomer (TPE), polyethylene terephthalate (PET), polytetrafluoroethylene (PTFE), glass, wood, and porcine skin) were all very high at ~370 J m$^{-2}$ (Fig. 4d, f). This is because, the gel point state with high viscoelasticity provided numerous PBA-rich dangling chains which generated strong van der Waals forces with the substrates. No significant difference in the adhesion strength was observed for different substrates, suggesting that the dominating fracture event occurred in the ultrasoft ionogel itself (see the inset photo of the peeling process in Fig. 4d). We also highlight that the P(BA-co-MAA) ionogel is totally hydrophobic and both the mechanical behavior and adhesion performance are resistant to water attack (Fig. 4e and f, Supplementary Fig. 17). The hydrophobic PBA chains and F-rich ionic liquid can help destroy the hydrated layer on the substrates, enabling strong adhesion by eliminating the interference of water molecules[48,49]. Even after immersing in water for 12 h, the underwater adhesion strength remained high at ~300 J m$^{-2}$ on various substrates, suggesting its potential uses in sweaty and aquatic environments.

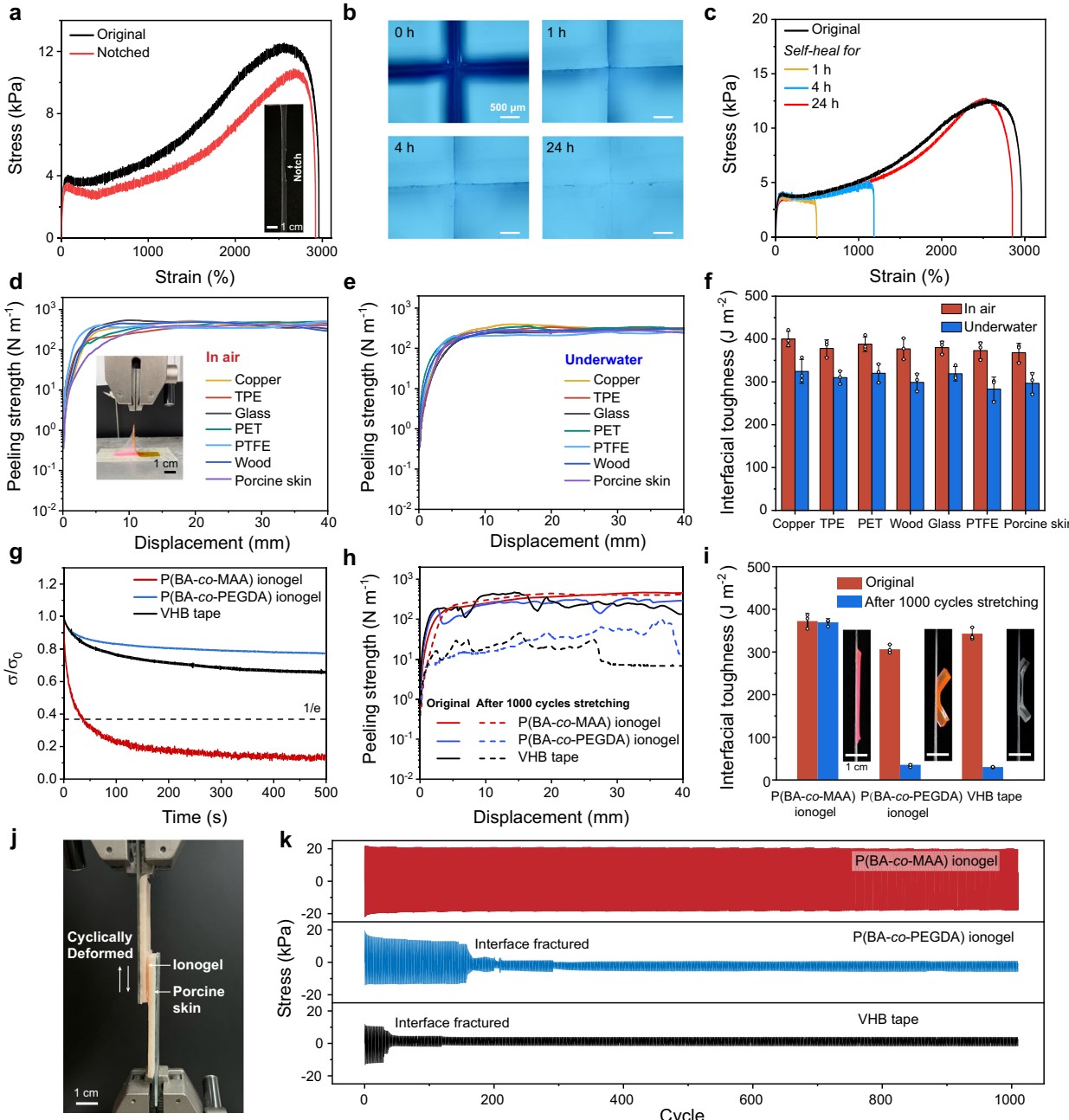

**Fig. 4 | Flaw-insensitive, self-healing, self-adhesive, and self-compliant properties of P(BA-co-MAA) ionogel. a** Tensile stress-strain curves of original and notched P(BA-co-MAA$_{0.05}$) ionogels. **b** Self-healing process of the ionogel captured by an optical microscope. **c** Tensile curves of the ionogel healed for different periods at room temperature. **d–f** 90° peeling curves and corresponding interfacial toughnesses of the ionogel with various substrates in air and underwater. The inset in (d) is a photo of the peeling process. **g** Stress relaxation curves of P(BA-co-MAA) ionogel, PBA-co-PEGDA ionogel, and VHB tape measured at 200% tensile strain. **h, i** 90° peeling curves and corresponding interfacial toughnesses of the three samples before and after 1000 cycles of TPE substrate stretching. **j** Experimental set-up for cyclic lap-shear tests (strain: ±5%) with the adhesive samples sandwiched between porcine skins. **k** Lap-shear stress curves during the repeated loading-unloading cycles until adhesion failure. Data are presented as the mean values ± SD, $n = 3$ independent samples. Source data are provided as a Source Data file.

Additionally, as mentioned before, the gel point state brings maximum stress relaxation and thus the best performance for durable self-compliance. To demonstrate the advantageous self-compliant behavior of P(BA-co-MAA) ionogel, chemically crosslinked PBA-co-PEGDA ionogel (PEGDA: polyethylene glycol diacrylate) and commercial VHB tape were selected as the control samples. The static adhesion strengths of PBA-co-PEGDA ionogel and VHB tape were carefully adjusted to be approximately equal to that of P(BA-co-MAA) ionogel for reasonable comparison. As shown in Fig. 4g, very rapid stress relaxation was observed for P(BA-co-MAA) ionogel with a relaxation time of merely 37 s, far smaller than our previously reported highly damping ionic skin and many biological tissues[19,25]. In contrast, both PBA-co-PEGDA ionogel and VHB tape are highly elastic with very slow stress relaxation (relaxation time »500 s). Such a difference was clearly demonstrated by the steel ball impact experiment, in which the free-falling ball strongly rebounded on the films of elastic PBA-co-PEGDA ionogel and VHB tape, but rapidly calmed down on the self-compliant P(BA-co-MAA) ionogel film (Supplementary Movie 2). Moreover, the

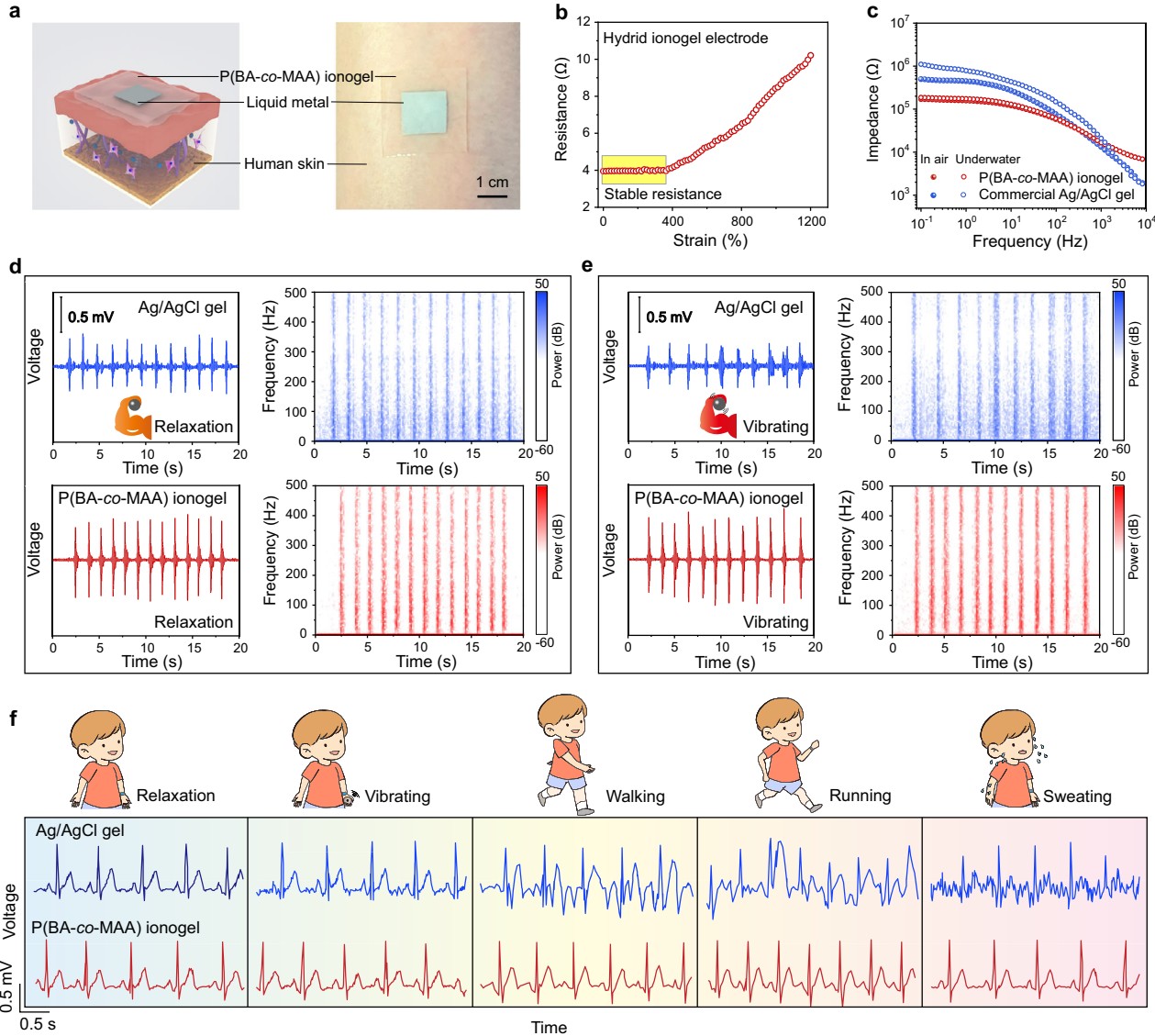

**Fig. 5 | Epidermal electronics applications. a** Schematic and photograph of epidermal P(BA-*co*-MAA)/liquid metal hybrid ionogel electrode. **b** Strain-dependent resistance changes of the hybrid ionogel electrode. **c** Skin interfacial impedances of P(BA-*co*-MAA) ionogel and commercial gel electrodes in air and underwater, respectively. **d, e** EMG pulses and corresponding FFT spectrogram analyses recorded with P(BA-*co*-MAA) ionogel and commercial gel electrodes at the relaxation and vibrating states, respectively. **f** ECG signals recorded with P(BA-*co*-MAA) ionogel and commercial gel electrodes under various exercise conditions.

gel-point-state behavior of P(BA-*co*-MAA) ionogel also facilitated its spontaneous penetration into the surface microstructures of PET substrate in an interlocked configuration (Supplementary Fig. 18). As a comparison, apparent gaps were observed in the cases of elastic PBA-*co*-PEGDA ionogel and VHB tape.

We then assessed the dynamic adhesion strengths of the three materials on a microstructured TPE substrate which was cyclically stretched to a fixed strain of 100%. As expected, the adhesion strength of P(BA-*co*-MAA) ionogel did not change even after 1000 cycles of stretching (Fig. 4h and i, and Supplementary Movie 3). However, the elastic PBA-*co*-PEGDA ionogel and VHB tape could not accommodate the dynamic deformation of TPE substrate, and a dramatic reduction of adhesion strength took place with remarkable adhesion failure. Similar results were also observed on a repeatedly inflated/deflated balloon (Supplementary Fig. 19). To evaluate the dynamic adhesion on natural skin, we further carried out the lap-shear tests by sandwiching the three adhesive materials between two porcine skin substrates (Fig. 4j). Cyclically shearing the three samples within ±5% strain produced dynamic stress curves that reflected the adhesion performance over time. Similar

to the peeling tests, the self-compliant P(BA-*co*-MAA) ionogel could sustain at least 1000 cycles of lap-shearing deformations, while the other two control samples failed within 150 cycles (Fig. 4k).

## Epidermal electronics applications

All the above combined properties of self-compliance, softness, adhesiveness, stretchability, self-healing, and flaw insensitivity highlight the advantages of self-compliant P(BA-*co*-MAA) ionogel over traditional viscous or elastic ionic skins when applied in epidermal electronics at dynamic conditions. Here we performed continuous electrophysiological monitoring as the demonstrative application, which is challenging to work at dynamic conditions due to the significant signal artifacts arising from frequent noises (e.g. movements, breathing, walking, tapping, running, sweating)[13]. We first examined the biocompatibility of P(BA-*co*-MAA) ionogel by the cytotoxicity tests on L929 cells which maintained >93% viability after 72 h (Supplementary Fig. 20). We also carried out the on-skin test of the ionogel, which caused no obvious irritation or injury to human skin in one day (Supplementary Fig. 21).

To circumvent the limit of stretchable ion-electron signal transduction, we additionally introduced a liquid metal/ionogel hybrid electrode (Fig. 5a). The thickness of the liquid metal layer was ~17 μm, and the particle size was 3-15 μm (Supplementary Fig. 22). Despite the introduction of liquid metal, the stress relaxation and rheological behavior of the whole epidermal electrode barely changed (Supplementary Figs. 23 and 24). The hybrid epidermal electrode has also very high conductivity ($2.3 \times 10^6$ S m$^{-1}$) as well as strain-insensitive resistance, favorable for stable signal transportation. The resistance remained unchanged till 360% strain (Fig. 5b), which is ascribed to the stretch-induced straightening of the tortuous conductive path of percolating liquid metal network[50]. In air, the interfacial impedance between human forearm skin and self-compliant P(BA-co-MAA) ionogel electrode was much lower than commercial Ag/AgCl gel electrode (Fig. 5c). By immersing in water, the skin interfacial impedance of commercial gel electrode significantly increased due to water invasion and gel swelling. In contrast, the interfacial impedance of P(BA-co-MAA) ionogel electrode was considerably stable owing to its hydrophobicity. All these results proved the promising capacity of P(BA-co-MAA) ionogel electrode for high-fidelity electrophysiological recording.

The epidermal electrodes were attached to the forearm for EMG measurement, which detected the action potentials induced by arm muscles. When the volunteer contracted back and forth musculus biceps brachii at the relaxation state, the self-compliant P(BA-co-MAA) ionogel electrode produced much stronger pulse signals than commercial gel electrode (Fig. 5d). Fast-Fourier transform (FFT) was performed to analyze the noise level in the EMG spectrogram. Clearly, less signal noise was observed in the ionogel electrode than in the commercial gel electrode. To explore the impact of skin vibrations on EMG monitoring, the same experiments were carried out when the volunteer held a vibrating ball (vibrating frequency ~100 Hz). More noise signals were also produced in the commercial gel electrode due to the unstable interfacial adhesion, while the ionogel electrode could still work well with an undisturbed signal-to-noise level (Fig. 5e).

The self-compliant ionogel electrode was further used to collect ECG signals in diversified exercise states (i.e., relaxation, vibrating, walking, running, and sweating). As demonstrated in Fig. 5f and Supplementary Fig. 25, stable ECG signals with almost unchanged signal-to-noise ratios (SNRs) were recorded in the case of ionogel electrode, while significantly reduced SNRs with severe motion artifacts were observed for the commercial gel electrode. All these results proved that the P(BA-co-MAA) ionogel electrode is suitable for both EMG and ECG recording in terms of reducing vibration/motion artifact interferences.

## Discussion

In this paper, we propose a supramolecular design for an epidermal ionic skin that achieves robust self-compliance with the underlying rough human skin. We highlight the effectiveness of phase-separated hierarchical hydrogen bond association, which enables the frequency-triggered release of longer polymer strands for enhanced topological chain entanglements that act as complementary crosslinks. Through careful recipe optimization, we demonstrate that the proof-of-concept P(BA-co-MAA) ionogel worked at the critical gel point state with almost equal viscosity and elasticity across a super-wide frequency range ($10^{-11}$ – 500 Hz), a phenomenon not observed in previous artificial ionic skins. The gel point state strikes a balance between dynamic compliance and shape recovery, contributing to the durable self-compliance of the ionogel on highly deformed substrates. Additionally, the ionogel exhibited natural skin-like softness, stretchability, self-adhesiveness, flaw-insensitivity, and self-healing properties, making it highly

suitable for the application in epidermal electronics. The ionogel-based hybrid electrode successfully acquired high-quality electrophysiological signals without motion artifact interference, demonstrating its robustness under dynamic conditions. We believe that the present design can be easily extended to other ionic materials for the development of more advanced epidermal electronic devices.

## Methods

### Materials

Butyl acrylate (BA, purity ~99%, catalog No. A0142), methyl acrylate (MA, purity ~99%, catalog No. A0145), ethyl acrylate (EA, purity ~99%, catalog No. A0143), isobutyl acrylate (iBA, purity ~99%, catalog No. A0747), and 2-methoxyethyl acrylate (MEA, purity ~98%, catalog No. A1405) were obtained from TCI (Shanghai). Methacrylic acid (MAA, purity ~98%, catalog No. M102642), bis(trifluoromethylsulfonyl)amine lithium salt (LiTFSI, purity ~99%, catalog No. B102576), and phenyl bis(2,4,6-trimethylbenzoyl)-phosphine oxide (BAPO, purity ~97%, catalog No. P138333) were purchased from Aladdin. Butyl-trimethylaminium chloride ($N_{4111}$Cl, purity ~98%, catalog No. R006631) was purchased from Rhawn. Eutectic gallium indium (EGaln, melting point ~16 °C) was purchased from Wochang Metal Co., Ltd. Poly(ethylene glycol) diacrylate (PEGDA, $M_n$ ~ 575 g mol$^{-1}$, purity ~100%, catalog No. 437441) was purchased from Sigma-Aldrich. All the liquid monomers and PEGDA were purified by passing through a basic alumina-filled column to remove inhibitor before use. Commercial Ag/AgCl gel electrode and VHB 4910 tape (thickness = 1 mm) were obtained from 3 M Company.

### Preparation of [$N_{4111}$][TFSI]

The ionic liquid, [$N_{4111}$][TFSI], was prepared according to the literature[48]. Briefly, 15.48 g of $N_{4111}$Cl was mixed with 100 mL of 1 M LiTFSI aqueous solution with vigorous stirring for 2 h. After phase separation, the lower oil layer was collected and washed with deionized water for five times. The resulting transparent ionic liquid of [$N_{4111}$][TFSI] was obtained after vacuum drying at 70 °C for 12 h.

### Preparation of P(BA-co-MAA) ionogel

The ionogel was synthesized by the random copolymerization of BA and MAA in the presence of [$N_{4111}$][TFSI] with tunable monomer ratios. The molar content of [$N_{4111}$][TFSI] was fixed to 0.1 with respect to BA. For the typical P(BA-co-MAA$_{0.05}$) ionogel, 10.35 g of BA (80 mmol), 0.35 g of MAA (4 mmol), 3.16 g of [$N_{4111}$][TFSI] (8 mmol), and 0.073 g of BAPO (0.17 mmol) were thoroughly mixed, and the mixed solution was then injected into a glass mold coated with release films. After polymerization with UV light (365 nm, 100 W) for 0.5 h under $N_2$ environment, the ionogel was finally obtained. Other ionogels (P(MA-co-MAA), P(EA-co-MAA), P(iBA-co-MAA), P(MEA-co-MAA), and P(BA-co-AA)) were prepared in the same procedure by simply replacing monomers or tuning feed ratios.

### Preparation of P(BA-co-MAANa) ionogel

The P(BA-co-MAA$_{0.05}$) elastomer was first prepared with the same procedure to the ionogel by using 10.35 g of BA (80 mmol), 0.35 g of MAA (4 mmol), and 0.073 g of BAPO (0.17 mmol). Subsequently, the elastomer was dissolved using a mixed solvent of methanol and toluene (volume ratio = 1:2) followed by the addition of 0.16 g of NaOH (4 mmol). The solution was stirred for 4 h until the ionization was completed. Then 3.16 g of [$N_{4111}$][TFSI] (8 mmol) was added to the solution and the solution was thoroughly mixed. Afterwards, the solution was poured into a mold and evaporated at room temperature for 12 h to remove most of the solvent. Finally, the P(BA-co-MAANa) ionogel was placed in a vacuum oven at 65 °C for 24 h to remove the residual solvent.

## Preparation of PBA-*co*-PEGDA ionogel

For the preparation of PBA-*co*-PEGDA ionogel, 10.35 g of BA (80 mmol), 0.023 g of PEGDA (0.04 mmol), 3.16 g of $[N_{4111}][TFSI]$ (8 mmol), and 0.073 g of BAPO (0.17 mmol) were mixed as the precursor. The above precursor was then injected into a glass mold coated with release films and UV cured for 0.5 h under $N_2$ environment to obtain the final PBA-*co*-PEGDA ionogel.

## Preparation of P(BA-*co*-MAA)/liquid metal hybrid ionogel electrode

First, 10 g of EGaIn was added into 2 mL of decanol and then sonicated in an ice water bath for 1 min with a probe sonicator (~300 W, Scientz, JY92-IIN) to prepare a liquid metal dispersion. The dispersion was screen-printed on a PET release film and baked in an oven at 60 °C for 1 h to remove the residual solvent. Afterwards, the precursor solution of P(BA-*co*-MAA$_{0.05}$) ionogel was poured into the mold consisting of the printed liquid metal pattern, and subsequent polymerization was initiated by exposing to UV light for 30 min. The hybrid ionogel electrode was finally peeled off from the mold (the conductive path of liquid metal was directly activated in this process). To avoid the risk of liquid metal leakage, a same thin ionogel layer was used to encapsulate the pattern.

## Preparation of microstructured substrates

To prepare the microstructured TPE substrate, 20 wt% solution of TPE in dichloromethane was poured onto a frosted glass mold and evaporated at room temperature for 12 h. Afterwards, the mold was dried in an oven at 50 °C for 24 h and the microstructured TPE film was obtained by peeling off from the mold. The microstructured PET substrate was prepared by laser engraving on a commercial PET film.

## Characterizations

The morphologies were examined by scanning electron microscope (SEM, Hitachi Regulus 8230) at an acceleration voltage of 3 kV. The surface nanostructure was captured by atomic force microscope (AFM, MFP-3D BIO, Oxford Instruments). The transparency of ionogel films (thickness ~500 μm) was evaluated using a UV-visible spectrophotometer (Lambda 950, Perkin Elmer). DSC measurements were performed on TA DSC250 at a scanning rate of 20 °C min$^{-1}$. The self-healing images were taken on an optical microscope (Olympus BX53-P). The rheological properties of the ionogel films (thickness ~1 mm) were investigated by a Thermo Scientific HAAKE MARS modular advanced rheometer using the 25 mm parallel-plate geometry. Tensile curves were recorded on a universal mechanical test machine (UTM2103, Shenzhen Suns technology) at ambient conditions. Electrochemical impedance measurements were conducted on an electrochemical workstation (CHI760E) by sandwiching polymer films in a symmetric stainless steel coin cell to evaluate ionic conductivities. The contact impedance of electrode-skin was measured by an electrochemical workstation (CHI760E) in the range from 0.1 Hz to 10 kHz with the applied voltage of 200 mV. The molecular weight of the copolymer in the ionogel was determined by gel permeation chromatography (GPC, Waters ACQUITY APC System) with tetrahydrofuran as the eluent.

## Small-angle X-ray scattering (SAXS)

SAXS experiments were performed using a laboratory-based SAXS-WAXS beamline, KWS-X (XENOCS XUESS 3.0 XL) at JCNS-MLZ, Garching, Germany. The MetalJet X-ray source (Excillum D2 + ) with a liquid metal anode was operated at 70 kV and 3.57 mA, emitting Ga-Kα radiation with a wavelength of λ = 1.314 Å. The sample-to-detector distances ranged from 0.5 to 1.7 m. The SAXS patterns were normalized to an absolute scale and azimuthally averaged to obtain the intensity profiles, and the empty cell background was subtracted. The 1D SAXS scattering curves were fitted by the 2-level Beaucage model to obtain the average radii of gyration. Temperature-dependent experiments were measured in a Kapton cell on a Peltier stage with temperature control.

## 1D/2D low-field NMR measurements

Low-field $^1H$ NMR tests were carried out on a VTMR20-010V-I NMR analyzer from Suzhou Niumag Analytical Instrument Corporation (magnet field strength: 0.5 $T$). A $^1H$ NMR probe was used to measure the $T_2$ profiles and 2D $T_1$-$T_2$ maps of P(BA-*co*-MAA) ionogels.

## Temperature-variable IR measurements

Temperature-variable IR spectra of the ionogel were collected in the transmission mode on a Nicolet iS50 FTIR spectrometer. Two ZnSe tablets were used to seal the sample and the temperature range was set from 20 to 200 °C with a ramp rate of 5 °C min$^{-1}$ (interval: 10 °C).

## Two-dimensional correlation spectroscopy (2DCOS)

The temperature-variable transmission IR spectra of the ionogel from 20 to 200 °C were used for performing 2D correlation analysis. The software 2D Shige ver. 1.3 (©Shigeaki Morita, Kwansei Gakuin University, Japan, 2004–2005) was used for 2D correlation analysis, and OriginPro program, ver. 10.0 was further used to draw contour maps. In 2DCOS spectra, red colors represent positive intensities, while blue colors represent negative intensities.

## Calculation of fractocohesive length

The fractocohesive length ($l_f$) was calculated using the formula $l_f = \Gamma/W$, where $\Gamma$ is the fracture energy, and $W$ is the work of fracture, which is calculated by integrating the stress-strain curve of the unnotched sample. The fracture energy was calculated in a pure shear configuration. Briefly, the notched and unnotched samples (gauge length of 10 mm, width of 5 mm, thickness of 1 mm, and/or precut of 1 mm) were both tested at the stretching speed of 50 mm min$^{-1}$. The fracture energy ($\Gamma$) was calculated from the integral area under the stress-strain curve of unnotched sample with the initial gauge length ($H$) according to the formula $\Gamma = H \times W(\varepsilon_c)$, where $\varepsilon_c$ is the critical strain at which fracture of notched sample occurred.

## 90° peeling test

The adhesion force was measured by the 90° peeling method with a vertical dynamometer (ESM303, MARK-10) at room temperature. Before the test, the ionogel (1 cm in width, 7 cm in length) with a polyimide film as the stiff back were adhered to different substrates under a small pressure. The samples were then transferred to different environments. The peeling speed was set to 50 mm min$^{-1}$.

## Cyclic lap-shear test

For cyclic lap-shear test, the adhesive materials (20 mm (length) × 25 mm (width) × 1 mm (thickness)) were sandwiched between two porcine skin (COFCO Joycome Foods) substrates. The tensile speed was set to 100 mm min$^{-1}$ and the strain was controlled to be ±5%.

## Cytotoxicity test

Cell viability tests of L929 cells (mouse fibroblast cell, purchased from ATCC, catalog No. CCL-1) incubated with different amounts of ionogel were performed by Cell-Counting-Kit-8 (CCK-8) assays. Before the preparation of the leaching solutions, the ionogel was sterilized by UV for 12 h. Then, the leaching solutions were obtained by soaking the sterilized ionogel at the concentrations of 2, 1.5, 1, and 0.5 mg mL$^{-1}$ in DMEM medium with 10% fetal bovine serum and 1% penicillin/streptomycin for 72 h at 37 °C. L929 cells were seeded in 96-well plates at a density of $1 \times 10^5$ cells per well, and cultured in a 5% $CO_2$ environment at 37 °C for 24 h. Subsequently, the leaching solutions of the ionogel were added to refresh the original culture medium. After co-culturing for 72 h, the original medium was replaced with the resulting DMEM. 100 μL of CCK-8 solution (diluted with DMEM at the ratio of 1:9) was

added to each well and incubated in the dark for 4 h at 37 °C. The absorbance was measured at 450 nm by a microplate reader.

### Electrophysiological monitoring measurements

EMG signals were measured in a triple-electrode configuration at a sampling frequency of 1000 Hz using a commercialized device (Sichiray Technology). Two working electrodes were attached to the flexor muscle of the left arm, separated by a distance of 2 cm. The reference (Ag/AgCl) gel electrode was placed at the elbow. For the detection of ECG signals, two working electrodes were attached to the insides of two wrists, and the reference (Ag/AgCl) gel electrode was attached to the left lower limb (ankle). ECG signals were measured by a Heal Force PC-80A ECG Monitor. A volunteer participated in the experiments as approved by the Institutional Biomedical Research Ethics Committee of Shanghai Institutes for Biological Sciences (No. 3011-1903).

### Statistics and reproducibility

All experiments were repeated independently with similar results for at least three times.

### Reporting summary

Further information on research design is available in the Nature Portfolio Reporting Summary linked to this article.

## Data availability

All data supporting the findings of this study are available within this article and Supplementary Information or from the corresponding authors upon request. The data generated in this study are provided in the Supplementary Information/Source Data file. Source data are provided with this paper.

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

## Acknowledgements

We gratefully acknowledge the financial supports from the National Natural Science Foundation of China (NSFC) (Nos. 22275032, 21991123, and 52322306 received by S.S.; No. 52161135102 received by P.W.). S.S. appreciates the supports from Shanghai Oriental Talent Program and Talent Development Fund. The authors also thank Zhenni Lu and Prof. Peng Wei of Donghua University for the assistance in cytotoxicity test.

## Author contributions

H.Y. carried out most experiments and co-wrote the manuscript. S.S. and P.W. supervised the project and co-wrote the manuscript. B.W. analyzed the SAXS results. All authors discussed the results and revised the manuscript.

## Competing interests

The authors declare no competing interests.

## Additional information

**Peer review information** : *Nature Communications* thanks Tao Chen, Yukikazu Takeoka and the other, anonymous, reviewer for their contribution to the peer review of this work. A peer review file is available.

