## [Peer Review File · Nature Communications]

Self-compliant ionic skin by leveraging hierarchical hydrogen bond associationREVIEWER COMMENTS

Reviewer #1 (Remarks to the Author):

In this work, Ye et al reported an interesting self-compliant ionic skin that works consistently at the gel point state over a super-wide frequency range. The proof-of-concept ionogel system was cleverly achieved by leveraging phase-separated hierarchical hydrogen bond association, which allows the frequency-triggered release of polymer strands to create topological entanglements as complementary crosslinks. The ionogel also exhibited natural skin-like softness, stretchability, self-adhesiveness, flaw-insensitivity, and self-healing properties, making it very suitable for the applications in epidermal electronics. In my opinion, the concept of this work is highly valuable in terms of both the basic science involved and the careful engineering behind it. I thus strongly recommend the publication of this work in Nature Communications. However, a minor revision may still be needed to further polish their scientific findings.

Comments:

1. The frequency-independent gel point state is the key to the ionogel's self-compliance. In consideration of the frequency-temperature equivalence, the rheological temperature-sweep curve of P(BA-co-MAA0.05) ionogel may also be provided for validating the judgement about gel point state across a super-wide frequency range.
2. In Supplementary Fig. 2, it is mentioned that the glass transition temperatures of the ionogels were not significantly influenced by the MAA content. Please provide a reasonable explanation.
3. What are the mechanical properties of the ionogel when the MAA content exceeds 0.3?
4. The frequency-sweep rheological curve of the elastomer without ionic liquids should be provided for comparison to show how ionic liquid affects the compliant properties of the ionogel.
5. The authors have compared the rheological curves of P(BA-co-MAA) ionogel electrodes with/without liquid metal to show that the hybrid ionogel is still self-compliant. I suggest adding the strain relaxation curve to further support this conclusion.

Reviewer #2 (Remarks to the Author):

This manuscript reported an interesting supramolecular design for an epidermal ionic skin which achieved robust self-compliance with the underlying rough human skin. The authors highlighted the role of phase separation with hierarchical hydrogen bond association, which enabled the frequency-triggered release of polymer strands for topological chain entanglements. The proof-of-concept P(BA-co-MAA) ionogel was demonstrated to work at the critical gel point state with almost equal viscosity and elasticity across a super-wide frequency range, which is very impressive and owns its novelty. The paper is well-written and supported by comprehensive data. With a few minor revisions, I believe that it is suitable for publication in Nature Communications.

1. The authors emphasized that the design for the gel-point-state materials is generalizable, and they also showed a few examples by using other monomers. The guiding rule for the selection of the two copolymerizing monomers may be further clarified or discussed.
2. The effect of ionic liquid on the rheological and stress relaxation behavior should be investigated by comparing the copolymer samples with and without ionic liquid.
3. What is the thickness of the ionogel film for rheological experiments? Will this parameter affect the rheological results?
4. The fracture energy should be provided for calculating the fractocohesive length (Fig. 4a).
5. I suggest adding more details about the liquid metal hybrid ionogel electrode, such as the thickness of the liquid metal layer, the particle size of liquid metal, as well as the mechanical properties of the final electrode. This will be informative for readers to reproduce this electrode.

Reviewer #3 (Remarks to the Author):

The authors found that a very simple material consisting of such as PBA-PMMA copolymer and an ionic liquid exhibits artificial skin-like properties.

This study is interesting because it notes that materials near the gel point can not only dissipate strain energy for dynamic compliance, but can also recover durably to avoid free flow.

Based on these findings, they developed a self-compliant ionic skin that works consistently at the critical gel point state with almost equal viscosity and elasticity across a super-wide frequency range (10¹⁰ – 500 Hz).

It is very comprehensive, with exhaustive experiments to optimize the properties of the materials.

Before this paper is accepted, please address the following points.

In Figure 1c.

the polymer chains are colored gray and light blue. In this study, do you consider gray to be PBA and light blue to be PMAA? However, since the synthesis method in this study is random copolymerization, it is unlikely that the chains consisting of block copolymers as shown in the figure are obtained. Figure 1c may be a convenient situation to explain the authors' idea, but it is not necessarily a diagram that explains what is happening in the system used in this study. a more accurate diagram should be made after investigating the copolymerizability of BA and MAA, etc.

In Figure 2c.

The size of the structure is assigned from the position of the peak that appears in the SAXS scattering profile, but what is observed in SAXS is not the size of the structure because of fluctuations.

Biocompatibility of Ionic Liquids and Ionic Gels

Explain in detail how the biocompatibility of the ionic liquids and ion gels used are being investigated using the L929 cell. 24 hour testing is being done, but this does not seem sufficient, so discuss

appropriate testing methods for the effects of these materials on the human body. Alternatively, explain the detailed examination of the biological effects of this ionic liquid, especially in the past, to determine if there is any toxicity in long-term use. Knowing this information is very important to the validity of what the authors have been working on.

Response to Reviewers

Reviewer #1:

In this work, Ye et al reported an interesting self-compliant ionic skin that works consistently at the gel point state over a super-wide frequency range. The proof-of-concept ionogel system was cleverly achieved by leveraging phase-separated hierarchical hydrogen bond association, which allows the frequency-triggered release of polymer strands to create topological entanglements as complementary crosslinks. The ionogel also exhibited natural skin-like softness, stretchability, self-adhesiveness, flaw-insensitivity, and self-healing properties, making it very suitable for the applications in epidermal electronics. In my opinion, the concept of this work is highly valuable in terms of both the basic science involved and the careful engineering behind it. I thus strongly recommend the publication of this work in Nature Communications. However, a minor revision may still be needed to further polish their scientific findings.

Response: We greatly thank the Reviewer's time and effort for evaluating our manuscript submitted to *Nature Communications* (NCOMMS-23-45121-T) as well as the very positive and constructive comments. According to the Reviewer's suggestions, we have carefully revised our manuscript. The following is the point-by-point response to all the questions. We hope that we have satisfactorily addressed all the Reviewer's concerns, and will be glad if the Reviewer could re-review our revised manuscript.

Comments:

1. The frequency-independent gel point state is the key to the ionogel's self-compliance. In consideration of the frequency-temperature equivalence, the rheological temperature-sweep curve of P(BA-co-MAA_{0.05}) ionogel may also be provided for validating the judgement about gel point state across a super-wide frequency range.

Response: We thank the respected Reviewer for this valuable suggestion. Indeed, the rheological temperature-sweep curve is helpful to validate our conclusion according to time-temperature superposition. In our revised manuscript, we have supplemented this data as Supplementary Fig. 8. The result shows that the G' and G'' curves overlapped in

a wide temperate range (10 - 200 °C), consolidating the presence of gel point state almost independent of frequency and temperature changes.

Corresponding revisions can be found in the highlighted text in Page 11 and Page S7.

2. In Supplementary Fig. 2, it is mentioned that the glass transition temperatures of the ionogels were not significantly influenced by the MAA content. Please provide a reasonable explanation.

Response: We thank the Reviewer for this comment. We ascribed the unnoticeable change of the glass transition temperatures of the ionogels with different MAA contents to the incompatibility of PMAA moieties with PBA and ionic liquid, which did not remarkably affect the segmental mobility of PBA-rich matrix at the studied compositions.

Corresponding revision can be found in the highlighted text in Page S3.

3. What are the mechanical properties of the ionogel when the MAA content exceeds 0.3?

Response: We thank the Reviewer for this comment. We have supplemented the tensile stress-strain curves with the MAA content increasing from 0.3 to 1 in Supplementary Fig. 4. As shown, along with the formation of bicontinuous structure, the ionogel will become stiffer and less ductile. A higher MAA content led to a high modulus and a short elongation.

Corresponding revisions can be found in the highlighted text in Page 10 and Page S4.

4. The frequency-sweep rheological curve of the elastomer without ionic liquids should be provided for comparison to show how ionic liquid affects the compliant properties of the ionogel.

Response: We thank the Reviewer for this constructive suggestion. The frequency-sweep rheological master curve of the P(BA-co-MAA_{0.05}) elastomer without ionic liquid has been supplemented in Supplementary Fig. 11. Similar to the ionogel, the elastomer

exhibited also the gel point state in a super-wide frequency range. This suggests that, ionic liquid did not significantly affect the compliant properties of the ionogel. The gel point state is mainly governed by the associating components, which control the whole chain dynamics.

Corresponding revisions can be found in the highlighted text in Page 12 and Page S9.

5. The authors have compared the rheological curves of P(BA-co-MAA) ionogel electrodes with/without liquid metal to show that the hybrid ionogel is still self-compliant. I suggest adding the strain relaxation curve to further support this conclusion.

Response: We appreciate the Reviewer for this valuable suggestion. As suggested, we have supplemented the strain relaxation curve of the hybrid ionogel, as shown in the supplemented Supplementary Fig. 23. No apparent changes in the stress relaxation behavior can be observed with the introduction of liquid metal, suggesting that liquid metal did not significantly affect the self-compliance of the hybrid ionogel.

Corresponding revision can be found in the highlighted text in Page 19 and Page S15.

Reviewer #2:

This manuscript reported an interesting supramolecular design for an epidermal ionic skin which achieved robust self-compliance with the underlying rough human skin. The authors highlighted the role of phase separation with hierarchical hydrogen bond association, which enabled the frequency-triggered release of polymer strands for topological chain entanglements. The proof-of-concept P(BA-co-MAA) ionogel was demonstrated to work at the critical gel point state with almost equal viscosity and elasticity across a super-wide frequency range, which is very impressive and owns its novelty. The paper is well-written and supported by comprehensive data. With a few minor revisions, I believe that it is suitable for publication in Nature Communications.

Response: We greatly thank the respected Reviewer's time and effort for evaluating our manuscript submitted to *Nature Communications* (NCOMMS-23-45121-T) as well

as the very positive and constructive comments. According to the Reviewer's suggestions, we have carefully revised our manuscript. The following is the point-by-point response to all the questions. We hope that we have satisfactorily addressed all the Reviewer's concerns, and will be glad if the Reviewer could re-review our revised manuscript.

1. The authors emphasized that the design for the gel-point-state materials is generalizable, and they also showed a few examples by using other monomers. The guiding rule for the selection of the two copolymerizing monomers may be further clarified or discussed.

Response: We thank the Reviewer for this constructive comment. In our work, we have shown a successful example of P(BA-co-MAA) ionogel as the gel-point-state material. We highlighted the importance of choosing an associative copolymer which contains two moieties with distinct affinities, and thus phase separation can occur with hierarchical H-bond assemblies acting as the frequency-sensitive physical crosslinks. To clarify the respective roles of different components, we further replaced BA by MA, EA, iBA, and MEA, which all showed the gel point state over a super-wide frequency range (Supplementary Fig. 10). If ionic liquid was removed from the ionogel, the resulting P(BA-co-MAA) elastomer exhibited still good gel point state in a super-wide frequency range (Supplementary Fig. 11, newly added). However, if we replaced MAA by AA or MAANa, the wide-frequency-range gel point state behavior cannot be well reproduced (Supplementary Fig. 12, newly added). All these control experiments suggest that, designing gel-point-state materials should mainly focus on the proper selection of associating components, which are the key to governing the whole chain dynamics. As we demonstrate in this paper, a phase-separated hierarchical H-bond with proper interaction strength can help achieve the frequency-independent gel point state. Corresponding descriptions have been added in the highlighted text in Page 12.

2. The effect of ionic liquid on the rheological and stress relaxation behavior should be

investigated by comparing the copolymer samples with and without ionic liquid.

Response: We thank the reviewer for this suggestion. As suggested, we have supplemented the rheological and stress relaxation data of the elastomer without ionic liquid in Supplementary Fig. 11. As stated in the last response, we found that the resulting P(BA-co-MAA) elastomer exhibited also the gel point state in a super-wide frequency range, as well good stress relaxation behavior, much similar to its ionogel. Corresponding revisions can be found in the highlighted text in Page 12 and Page S9.

3. What is the thickness of the ionogel film for rheological experiments? Will this parameter affect the rheological results?

Response: We thank the Reviewer for this comment. The thickness of the used ionogel film for rheological experiments was 1 mm. We have also compared the frequency-sweep rheological curves of the ionogel films with different thicknesses. As shown in **Figure R1**, no significant changes in both G' and G'' curves and $\tan \delta$ values (at 0.1 Hz) were observed for these samples. This suggests that, the film thickness does not apparently affect the self-compliance of the ionogel.

Figure R1. Rheological curves and $\tan \delta$ values (at 0.1 Hz) of P(BA-co-MAA) ionogels with different thicknesses.

4. The fracture energy should be provided for calculating the fractocohesive length (Fig. 4a).

Response: We thank the Reviewer for this suggestion. The fracture energy has been provided in the highlighted text in Page 15.

5. I suggest adding more details about the liquid metal hybrid ionogel electrode, such as the thickness of the liquid metal layer, the particle size of liquid metal, as well as the mechanical properties of the final electrode. This will be informative for readers to reproduce this electrode.

Response: We greatly appreciate the Reviewer for this suggestion. In our revised manuscript, we have supplemented the thickness of liquid metal layer and the particle size characterized by SEM (Supplementary Fig. 22, newly added). The thickness of the liquid metal layer was $\sim 17 \mu\text{m}$, and the particle size was 3-15 μm . Besides, the stress relaxation behavior of the hybrid electrode has been provided in the supplemented Supplementary Fig. 23. The introduction of liquid metal did not significantly affect the mechanical properties of P(BA-co-MAA) ionogel.

Corresponding revisions can be found in the highlighted text in Page 19 and Pages S14, S15.

Reviewer #3:

The authors found that a very simple material consisting of such as PBA-PMMA copolymer and an ionic liquid exhibits artificial skin-like properties.

This study is interesting because it notes that materials near the gel point can not only dissipate strain energy for dynamic compliance, but can also recover durably to avoid free flow.

Based on these findings, they developed a self-compliant ionic skin that works consistently at the critical gel point state with almost equal viscosity and elasticity across a super-wide frequency range ($10^{-11} - 500 \text{ Hz}$).

It is very comprehensive, with exhaustive experiments to optimize the properties of the materials. Before this paper is accepted, please address the following points.

Response: We greatly thank the respected Reviewer's time and effort for evaluating

our manuscript submitted to *Nature Communications* (NCOMMS-23-45121-T) as well as the positive and constructive comments. According to the Reviewer's suggestions, we have carefully revised our manuscript. The following is the point-by-point response to all the questions. We hope that we have satisfactorily addressed all the Reviewer's concerns, and will be glad if the Reviewer could re-review our revised manuscript.

1. In Figure 1c, the polymer chains are colored gray and light blue. In this study, do you consider gray to be PBA and light blue to be PMAA? However, since the synthesis method in this study is random copolymerization, it is unlikely that the chains consisting of block copolymers as shown in the figure are obtained. Figure 1c may be a convenient situation to explain the authors' idea, but it is not necessarily a diagram that explains what is happening in the system used in this study. A more accurate diagram should be made after investigating the copolymerizability of BA and MAA, etc.

Response: We gratefully appreciate the Reviewer for this valuable comment. Yes, in Figure 1c, the gray chains denote PBA and the light blue chains denote PMAA. We fully agree with the Reviewer that the synthesis method in our study is random copolymerization. This is supported by the reactivity ratios of BA and MAA ($r_{\text{BA}} = 0.35$, $r_{\text{MAA}} = 1.31$, $r_{\text{BA}} \times r_{\text{MAA}} < 1$) (Ref. 34: Paxton, T. R. *J. Polym. Sci., Part B: Polym. Lett.* 1963, 1, 73). Therefore, the phase-separated domain should be PMAA-rich phase, instead of neat PMAA phase. We apologize for the misleading scheme in the original Figure 1c. In the revised manuscript, we have updated this diagram by adding some PBA repeating units in the phase-separated domains.

Corresponding revisions can be found in Page 7, 30-31, and 34.

2. In Figure 2c, the size of the structure is assigned from the position of the peak that appears in the SAXS scattering profile, but what is observed in SAXS is not the size of the structure because of fluctuations.

Response: We thank the Reviewer for this valuable comment. In a monodisperse dilute system, the scattering behavior typically exhibits Guinier scattering within a small

range of measurable q values. One can observe plateaus in the small q region of a double logarithmic coordinate plot. In cases of systems with high concentrations, there may be the presence of a correlation peak in the small q region. However, the system we have studied involves multi-scale structural scatterers, featuring hierarchical structures with significant differences in size within the sample. Given that the scattering intensity of larger structures has a more pronounced impact on scattering than smaller structures, in a multi-scale structural system, the scattering of small-scale structures is influenced by larger scatterers. This leads to the absence of plateaus but rather a subtle bump, as depicted in **Figure R2** (same to Fig. 2c, on a log-log scale). Upon closer examination, it becomes evident that there are discernible changes in the scattering curve that correspond to the feedback effect of the scatterer size within the studied scale range. Consequently, we employed a multi-scale Beaucage model to fit the scattering curve and deduce the sizes of the scatterers present in the sample (Beaucage model: *J. Appl. Crystallogr.* 1995, 28, 717).

Figure R2. SAXS profiles of PBA-co-MAA ionogels with increasing MAA contents from 0 to 0.3. The dash lines and the arrow highlight the presence of phase separation in the scattering curves.

3. Biocompatibility of Ionic Liquids and Ionic Gels.

Explain in detail how the biocompatibility of the ionic liquids and ion gels used are

being investigated using the L929 cell. 24 hour testing is being done, but this does not seem sufficient, so discuss appropriate testing methods for the effects of these materials on the human body. Alternatively, explain the detailed examination of the biological effects of this ionic liquid, especially in the past, to determine if there is any toxicity in long-term use. Knowing this information is very important to the validity of what the authors have been working on.

Response: We thank the Reviewer for this suggestion. We have added more details for the cytotoxicity test in the Methods (highlighted in Page 26-27). Indeed, 24 h testing may not be sufficient, and in the revised manuscript, we have extended the cell incubation time to 72 h, as shown in the supplemented Supplementary Fig. 20. Since there were almost no water-soluble masses in the ionogel, even in 72 h, the ionogel still exhibited excellent biocompatibility with cell viabilities larger than 93%.

We also totally agree with the Reviewer that the biocompatibility and toxicity information is very important for on-skin electronics applications. This is also why we chose N₄₁₁₁TFSI as the ionic liquid for preparing the self-compliant ionogel. As we reported previously (ref. 16: Wu, et al. *Adv. Funct. Mater.* 2021, 31, 2107226), N₄₁₁₁TFSI is a low-toxic ionic liquid as evidenced by in vitro mouse model test. This is also supported by another study (de la Fuente-Nunez, et al. *ACS Nano* 2021, 15, 966), which showed that the ammonium:TFSI ionic liquid had anti-infective activity with no evident toxic side effects, superior to imidazolium-based ionic liquids. To test the toxicity of our ionogel to human body, we additionally carried out an on-skin irritation test, as shown in the supplemented Supplementary Fig. 21. There was no obvious irritation or injury to human skin in 24 h, suggesting that the ionogel is skin-friendly. However, since the ionogel is not breathable, long-time attachment on skin may not be recommended.

Corresponding revisions can be found in the highlighted text in Pages 19, 26, 27, 29, and Pages S13, S14.

REVIEWERS' COMMENTS

Reviewer #1 (Remarks to the Author):

The current version after revision could be accepted as is.

Reviewer #2 (Remarks to the Author):

The authors have well addressed all the issues raised. This work can be considered for publication without further revision.

Reviewer #3 (Remarks to the Author):

I am pleased with Authors' courteous response and their revision of the paper. I heartily recommend publishing this paper in Nature Communications.